

# *pathVar:* a new method for pathway-based interpretation of gene expression variability

Laurence de Torrente[1], Samuel Zimmerman[1], Deanne Taylor[2,3], Yu Hasegawa[1], Christine A. Wells[4] and Jessica C. Mar[1,5,6]

[1] Department of Systems and Computational Biology, Albert Einstein College of Medicine, Bronx, NY, United States of America
[2] Department of Biomedical and Health Informatics, Children's Hospital of Philadelphia, Philadelphia, PA, United States of America
[3] Department of Pediatrics, University of Pennsylvania, Philadelphia, PA, United States of America
[4] Department of Anatomy and Neuroscience, University of Melbourne, Melbourne, Victoria, Australia
[5] Department of Epidemiology and Population Health, Albert Einstein College of Medicine, Bronx, NY, United States of America
[6] University of Queensland, Australian Institute for Bioengineering and Nanotechnology, Brisbane, Queensland, Australia

Corresponding author
Jessica C. Mar,
jessica.mar@einstein.yu.edu

## ABSTRACT

Identifying the pathways that control a cellular phenotype is the first step to building a mechanistic model. Recent examples in developmental biology, cancer genomics, and neurological disease have demonstrated how changes in the variability of gene expression can highlight important genes that are under different degrees of regulatory control. Simple statistical tests exist to identify differentially-variable genes; however, methods for investigating how changes in gene expression variability in the context of pathways and gene sets are under-explored. Here we present *pathVar,* a new method that provides functional interpretation of gene expression variability changes at the level of pathways and gene sets. *pathVar* is based on a multinomial exact test, or an asymptotic Chi-squared test as a more computationally-efficient alternative. The method can be used for gene expression studies from any technology platform in all biological settings either with a single phenotypic group, or two-group comparisons. To demonstrate its utility, we applied the method to a diverse set of diseases, species and samples. Results from *pathVar* are benchmarked against analyses based on average expression and two methods of GSEA, and demonstrate that analyses using both statistics are useful for understanding transcriptional regulation. We also provide recommendations for the choice of variability statistic that have been informed through analyses on simulations and real data. Based on the datasets selected, we show how *pathVar* can be used to gain insight into expression variability of single cell versus bulk samples, different stem cell populations, and cancer versus normal tissue comparisons.

## INTRODUCTION

Global studies of gene expression provide two quantitative parameters: a commonly-used metric is the relative abundance of a transcript (and group differences in transcript abundance), likewise the expression variability of that transcript provides insight into the heterogeneity of a sample group (*Mason et al., 2014*), and expression variability changes between groups have been shown to reflect underlying changes in transcriptional regulatory processes (*Blake et al., 2003*; *Chalancon et al., 2012*; *Munsky, Neuert & Van Oudenaarden, 2012*; *Raser & O'Shea, 2004*). Patterns of variability in gene expression have provided insight into how pathways are regulated in cells (*Burga, Casanueva & Lehner, 2011*; *Raj et al., 2010*); especially in the context of single cell profiling studies, where the average expression of a gene in a cell population carries limited information for understanding transcriptional regulation. Recent studies have identified pathways showing differential control or regulatory constraint that were discovered only by modeling changes in gene expression variability and were not apparent from standard analyses of average gene expression (*Yu et al., 2008*). While variability is becoming more prevalent as an informative metric, the current challenge lies in how to interpret these analyses to maximize functional information, such as with respect to pathways and curated gene sets. Due to the newness of this area, statistical methods for investigating expression variability are currently under-developed, and lacking for pathway-centric approaches. It is necessary, therefore, to develop such methods since information on expression variability can be used to complement analyses of average expression, and improve our understanding of the transcriptional state of the cell.

Intuitively, the distribution of gene expression variability in a pathway highlights the subset of genes with different degrees of regulatory control (Fig. 1). In the case of a one-group design, where multiple profiles represent replicates of the same phenotype, e.g., different embryonic stem cell (ESC) lines, identifying pathways that have an unexpected proportion of low variability genes may point to those that contribute integral roles for stem maintenance or regulation (*Mason et al., 2014*). To appreciate this, consider two previous studies that provided evidence linking criticality of genes and their decreased variability in expression. One study (*Yu et al., 2008*) identified genes with decreased expression variability in tumors relative to normal tissue; this gene set, termed the Posed Gene Cassette, included key genes whose expression impacted metastasis and patient survival as demonstrated through *in vivo* and *in vitro* experimental approaches. More recently, a second study (*Hasegawa et al., 2015*) showed that genes with decreased variability in expression for four stages of early embryonic development (4-cell, 8-cell, morula and blastocyst) were more likely to be associated with essentiality, haploinsufficiency or ubiquitous expression, suggesting that these stably-expressed genes contribute to cell survival.

In the two-group design, where profiles are compared between two contrasting phenotypes, e.g., ESCs versus induced pluripotent stem cells (iPSCs), identifying pathways associated with different patterns of expression variability may highlight those pathways that contribute to group-specific differences. Previous studies have analyzed the enrichment of genes with different levels of expression variability for specific pathways (*Hasegawa et al., 2015*; *Mar et al., 2011*); however, these analyses are based on gene lists defined by an

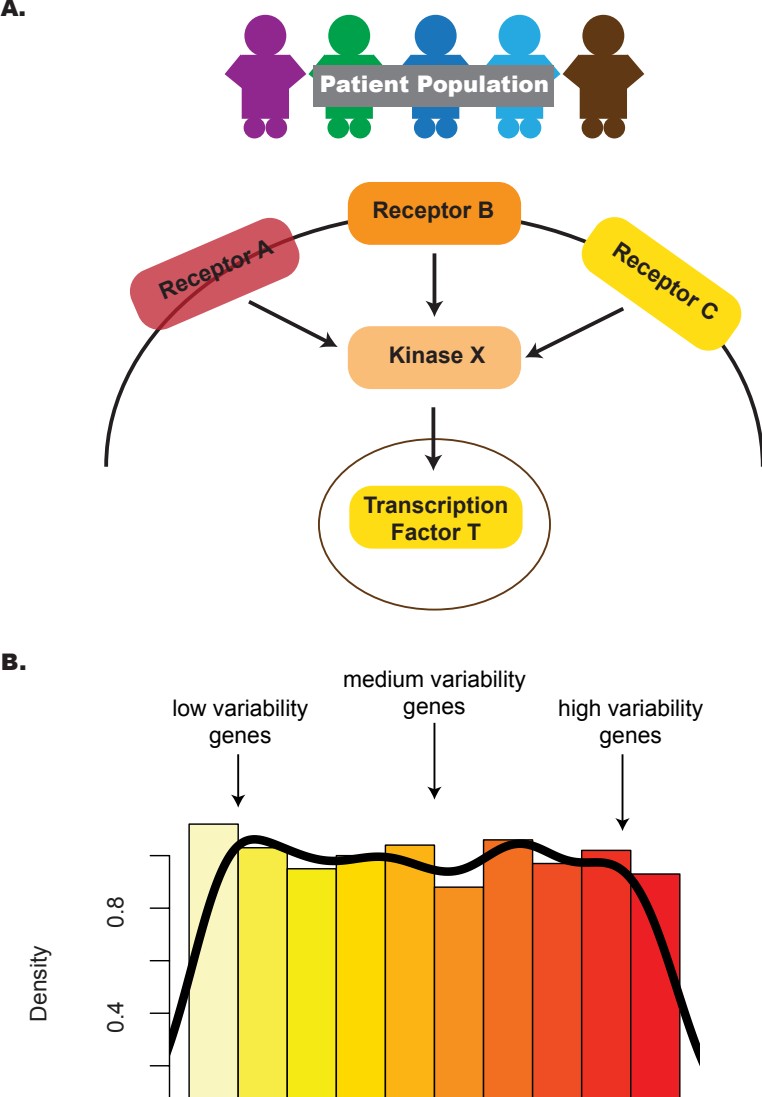

**Figure 1** **The distribution of gene expression variability highlights the regulatory control that different genes in the pathway are subjected to.** (A) Absolute gene expression is a proxy for how genes are transcriptionally regulated between samples. Studying the consistency of how genes are expressed can also add information on pathway control e.g., lower levels of inter-individual variability may reflect increased regulatory control. (B) By considering the distribution of gene expression variability, we may be able to understand transcriptional regulation in a more comprehensive manner—this is the premise of the *pathVar* method.

arbitrary cut-off and do not take into account the expression distribution of genes in the pathway. One would expect that more informative results could be obtained by focusing on the shape of the expression distribution in a statistically rigorous manner, much like a gene set enrichment analysis (GSEA) (*Mootha et al., 2003*) analogue for variability instead of relying only on average expression, or over-representation (OR) analyses (*Falcon & Gentleman, 2007*). Computational methods to implement these kinds of approaches are currently lacking for expression variability.

Our method, *pathVar,* addresses this gap by providing a pathway-based analysis of gene expression variability where pathways are assessed based on deviations of their gene expression variability distribution relative to a reference. In the one-group setting, the reference can be the global distribution constructed from all genes. In a two-group setting, one of the groups serves as the reference or control group. For each pathway, our method also identifies which genes in a pathway show aberrant levels of gene expression variability (Fig. 1). Additionally, we provide guidance on selecting an appropriate variability statistic based on analyses of simulated and real data. We also highlight how *pathVar* can be used to further understand transcriptional regulation based on gene expression variability for selected group comparisons, e.g., single cell versus bulk data, different stem cell lines, and cancer versus normal tissue.

## METHODS

The *pathVar* method can be summarized in three main steps (Fig. 2).

### Step 1: selecting a statistical measure to estimate gene expression variability

Variability is defined as the amount of dispersion in a given distribution (*Larsen & Marx, 2017*). Different statistical measures are available to estimate gene expression variability, and in genomics, the estimators that are most often employed are the standard deviation (SD) (Eq. (1)) (*Hasegawa et al., 2015*), the coefficient of variation (CV) (Eq. (2)) (*Mar et al., 2011*; *Mason et al., 2014*), and the median absolute deviation (MAD (Eq. (3)) (*Wijetunga et al., 2014*). Conceptually, these statistics share similarities in their mathematical definition, and each one comes with their own advantages, as is often the case with any estimator that is applied to data.

Let $x_1, \ldots, x_n$ denote a univariate dataset, then

$$SD = \sqrt{\frac{1}{n-1}\sum_{i=1}^{n}(x_i - \overline{x})^2} \tag{1}$$

$$CV = \frac{SD}{\overline{x}} \tag{2}$$

$$MAD = median(|x_i - median(x)|). \tag{3}$$

A consensus on which estimator should be adopted for variability analysis remains unclear, and this is partly because performance of the estimators appears to be data-specific. The SD is often the preferred estimator for measuring gene expression variability since it is on

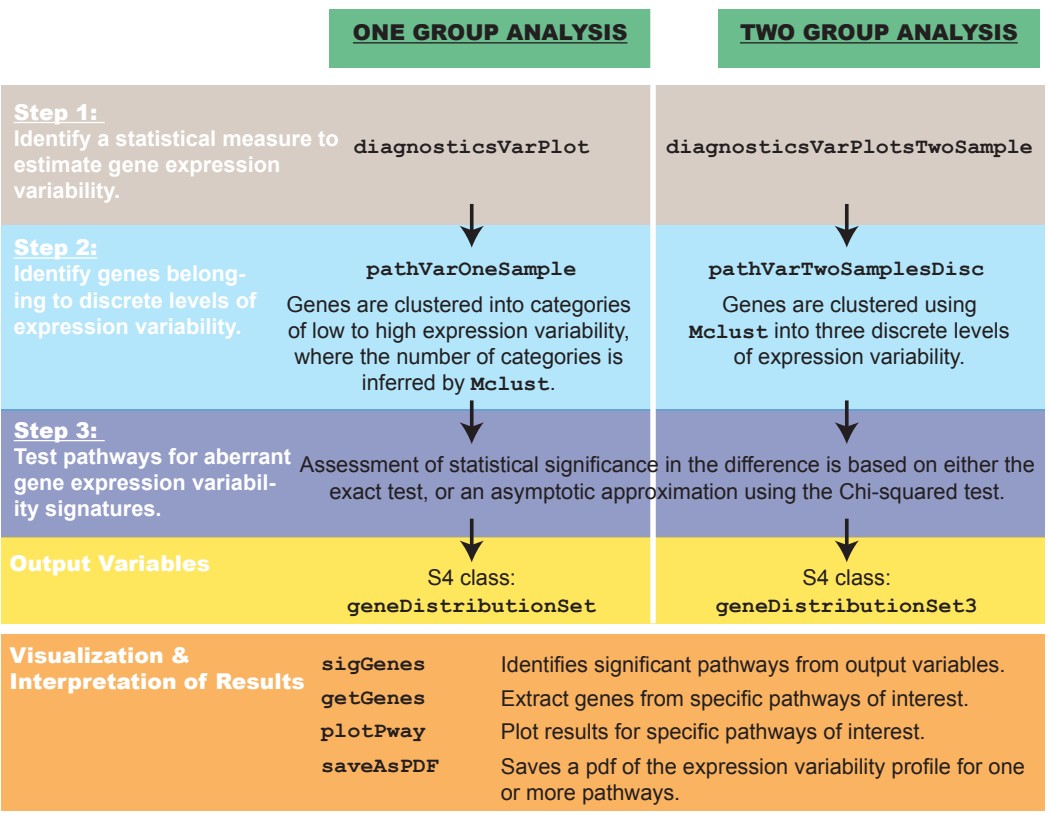

**Figure 2** Overview of *pathVar*, including the main functions in the R package.

the same scale as the average and therefore easy to interpret. The variance (i.e., $SD^2$) is also characterized by the second central moment of a distribution, and hence the SD is directly linked to one of the fundamental metrics characterizing the probability distribution. A criticism of the SD, however, is that it may be dependent on average expression and therefore it is necessary to investigate the association between these two measures. To address this concern, the CV, which represents the ratio of the SD and the average, is often used; however it also has its own drawbacks. Most significantly, it can be affected by zero-inflation which occurs when near zero levels of average expression result in extremely large values of CV that do not necessarily reflect a large degree of overall variability. The MAD is a robust measurement of dispersion that behaves well in the presence of outlier data points. It has previously been used to study DNA methylation variability (*Wijetunga et al., 2014*).

Data simulations suggest that in general, the SD shows stronger performance as the variability estimator compared to MAD and CV in the *pathVar* method (see Text S1). The simulations were conducted to test the performance of each estimator under a wide variety of conditions. The choice of an estimator can also be motivated by the expectation that an ideal estimator of gene expression variability will be uncorrelated with average gene expression. If the variability estimator is highly correlated with the average expression, then trends observed for expression variability may simply be recapitulated by those observed

for average expression. We therefore desire an estimator of variability that is the least correlated with average gene expression for an analysis of gene expression variability to be maximally informative. Our suggestion is to use the SD since overall this estimator displayed stronger performance in simulations or the estimator with the lowest correlation with average expression; however, ultimately, the final choice of the estimator is left to the user and can be easily specified in the software.

## Step 2: identifying genes that belong to categories of high, medium or low levels of expression variability

Using a specified estimator, all genes are assigned to a discrete level of expression variability. In the one-group case, assignments are based on clustering the data using Normal mixture models via the *mclust* algorithm (*Fraley & Raftery, 2002*). The number of clusters or mixtures corresponds to the discrete levels of variability and is a parameter inferred by *mclust*. The *mclust* algorithm considers a finite range of values (starting with a minimum of one level to a maximum of four by default) and chooses the number that is most appropriate for the data using the Bayesian Information Criterion. The upper limit of four levels is recommended out of simplicity, where it is more useful to model a handful of variability levels, e.g., low, medium, high and very high, whereas for much larger numbers, the interpretation ceases to be as intuitive. In the *pathVar* package, the user is, however, free to use whatever upper limit is appropriate for the analysis.

For the two-group case, assignments are based instead on the 33rd and 66th percentiles that are computed from the combined gene expression variability distribution of all genes from both groups in the dataset. Low variability genes are defined as those with values falling between 0 and the 33rd percentile, medium variability genes are those between the 33rd percentile and the 66th percentile, and high variability genes being greater than the 66th percentile. The variability levels are defined by a fixed number of standardized percentiles, instead of inferred as in the one-group case, because it is possible that a different number of variability levels might be inferred for the two groups. To ensure a straightforward and balanced comparison between the two groups, the number of discrete categories are fixed and the boundaries for variability levels are based on percentiles calculated from all the data. Under both the one-group and two-group cases, the outcome of Step 2 is to identify the fixed boundaries that define each of the discrete levels of expression variability.

## Step 3: testing pathways for aberrant gene expression variability signatures

The *pathVar* method decomposes each pathway into a set of counts corresponding to the number of genes in each discrete level of expression variability. By default, pathways from the Kyoto Encyclopedia of Genes and Genomes (KEGG) (*Kanehisa & Goto, 2000*) and REACTOME (*Croft et al., 2014*) are used; however, users may also import their own definitions. For a specific pathway, let $O_i$ denote the observed count of genes annotated to this pathway with expression variability in the $i$th level, where $i = 1, \ldots, m$ discrete levels. The total number of genes $n$ is defined as $n = \sum_{i=1}^{m} O_i$. A statistical test is used to evaluate whether the set of counts $(O_1, \ldots, O_m)$ associated with a pathway deviates significantly from either a reference count distribution in the one-group case, or between the two phenotypes

in the two-group case. In the one-group case, the reference distribution is obtained by counting the total number of genes in each level of gene expression variability (Fig. S1).

The null hypothesis that *pathVar* evaluates is:

- $H_{0(1)}$: the expression variability counts observed for a specific pathway were generated under the same distribution as the reference counts for the one-group case.
- $H_{0(2)}$: the variability-based counts for both groups were drawn from the same underlying distribution, for the two-group case.

To assess the deviation observed between the variability count distribution and an expected distribution for a specific pathway, we provide two statistical tests as options available for this analysis. Option 1 is the multinomial exact test, and Option 2 is the Chi-squared test.

For Option 1, the exact test models the counts as a multinomial random variable $(O_1, \ldots, O_m) \sim \text{Multi}(n, p)$ where $p = (p_1, \ldots, p_m)$ and $p_i$ is the probability that a gene belongs to the $i$th variability level for $i = 1, \ldots, m$ in the reference distribution. Under the exact test, the $P$-value is obtained by summing over all possible events that are less likely than the set of counts observed. If the $P$-value falls below a specified significance threshold (e.g., $P$-value $< 0.05$), sufficient evidence exists to reject the null hypothesis and we conclude that the pathway has an aberrant distribution of expression variability counts. In the one-group case, this means that the variability count distribution of the specific pathway deviates from the reference distribution i.e., all genes surveyed. In the two-group case, this result means that the variability is not identically distributed between the two contrasting phenotypes.

While the exact test is attractive because it calculates the $P$-value exactly, from a practical perspective, these kinds of tests can often be time and memory-intensive for genomic data, especially as the number of levels $m$ and the number of genes in the pathway $n$ grows. For example, consider a pathway with 30 genes, where the number of possible sets of counts to consider with three variability levels for the calculation of the $P$-value is 496. If the size of the pathway increases to 100, then the number of possibilities to consider grows to 5,151. From this basic example, it can be seen how increasing $n$ or $m$ can lead to some very extensive calculations.

Option 2 overcomes this limitation, as it is less computationally intensive, and tests the same null hypotheses using the Chi-squared test as an alternative to the exact test. The test statistic $X^2 = \sum_{i=1}^{m} \frac{(O_i - E_i)^2}{E_i}$ follows a Chi-squared distribution $X^2 \sim \chi^2_{m-1}$ with $m - 1$ degrees of freedom: $X^2 \sim \chi^2_{m-1}$. The expected counts $E_i = n * p_i$ are the expected number of genes in each level of expression variability within a specific pathway. The Chi-squared test achieves its computational efficiency because it is based on an asymptotic approximation, where the test becomes more accurate as $n$, the total number of genes in a pathway increases.

Both the exact test and the Chi-squared test assess whether a pathway has a significant change in gene expression variability. The resulting $P$-values from all pathways tested are adjusted using the Benjamini–Hochberg method which is based on controlling the false discovery rate (*Hochberg & Benjamini, 1990*). Finally, a pathway of interest can be

investigated further using a Binomial test to assess within each level of variability, whether a significant deviation exists between either the pathway and the reference in the one-group case, or between the two phenotypic groups. This pairwise difference for each independent level provides a means to pinpoint the subset of genes within a pathway showing deviation in expression variability.

### Power calculations indicate that *pathVar* has greater power to detect changes for pathways with skewed expression variability distributions than symmetric ones

Simulations were designed to investigate how experimental parameters influence the power of *pathVar* (Text S1). The results demonstrate that for a fixed effect size, the power of the Chi-squared test increases when the number of genes in a pathway also increases. Similarly, for a pathway of fixed size, the power of the test is higher for a larger effect size (Text S1, Fig. S1.5). The simulations also showed that the Chi-squared test had more power to detect differences between the reference distribution and a pathway that has a skewed variability distribution compared to a symmetric one. As an example, consider pathway $p_1$ where an effect size of 0.24 results in 80% power for approximately 150 genes in the pathway. For the symmetric distribution in pathway $p_2$, in order to obtain the same level of power, a pathway size of at least 200 genes is required (Text S1, Fig. S1.6).

The influence of sample size was also investigated on the power of the Chi-squared test for fixed pathway sizes (Text S1, Fig. S1.7). Higher levels of power were obtained for larger effect sizes, and this trend was more pronounced with smaller numbers of genes. As expected, the asymptote to 100% power occurred with smaller numbers of samples when the effect size was larger (0.5 or 0.9 versus 0.1). Overall, these calculations support the recommendation that experimental designs with larger samples will yield higher powered analyses, especially when the effect size is large.

## RESULTS

### Application of *pathVar* to human embryonic stem cell datasets identify significant pathways with distinct profiles of gene expression variability

To demonstrate the utility of *pathVar* in practice, the method was applied to three gene expression datasets of human ESCs (Text S2). The Bock dataset (*Bock et al., 2011*) had twenty ESC samples that were generated using microarray profiling, and the Yan dataset (*Yan et al., 2013*) had hESC samples profiled using single cell RNA-sequencing (RNA-seq) for eight cells at passage 0 (p0), and 26 cells at passage 10 (p10). *pathVar* was run independently on the three datasets, and pathways with a statistically significant deviation in their gene expression variability profile relative to the reference distribution were detected (Table S1, *P*-value < 0.01). Significant KEGG pathways reflected aberrant gene expression counts in ribosomes, metabolism (oxidative phosphorylation), the spliceosome and neurodegenerative pathways (Alzheimer, Parkinson and Huntington) (Tables S2A–S4A). Significant REACTOME pathways fell into three main classes representing cell cycle, metabolism and infectious disease (Tables S2B–S4B). Considerable overlap was observed

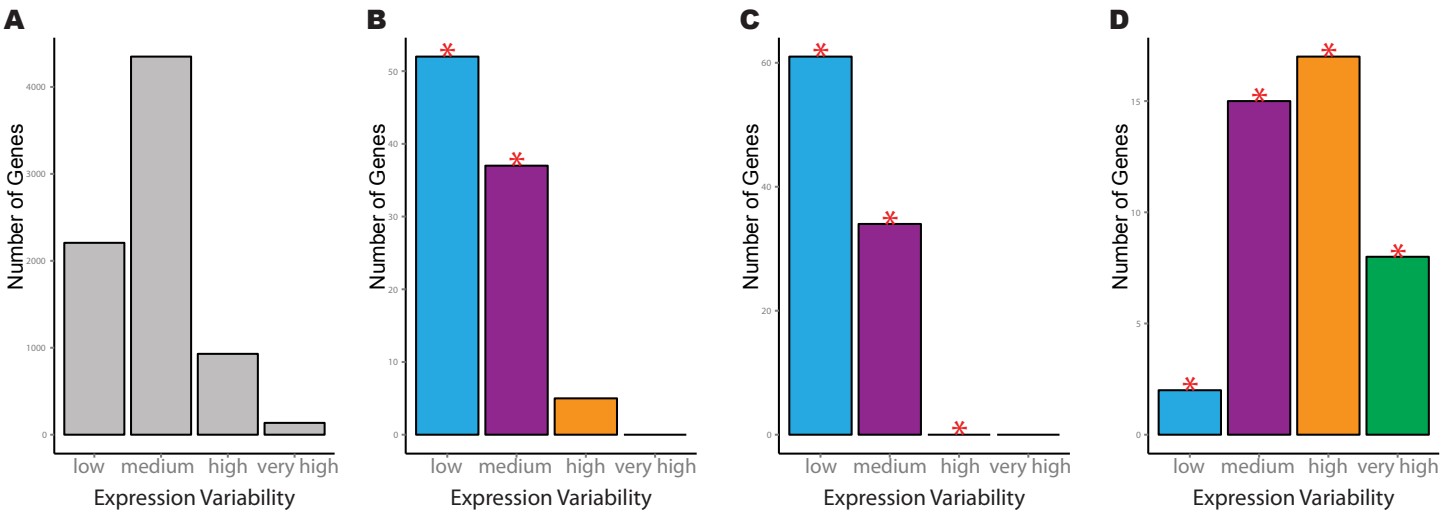

**Figure 3** **Example of four significant KEGG pathways for one-group *pathVar* analysis of the Bock embryonic stem cell data.** (A) Variability count distribution for the reference. (B) Splicesome pathway (hsa03040), (C) oxidative phosphorylation (hsa00190), (D) ECM-receptor interaction (hsa04512). The red stars indicate a significant difference between the pathway and reference distribution for a specific level of expression variability.

in the significant results obtained between the three datasets (Fig. S2), where cell cycle was the most highly represented REACTOME category. Similar percentage distributions were observed (Fig. S2), demonstrating consistency in the results obtained from *pathVar* despite differences in technology platforms, ESC lines and passage number.

We next inspect the variability count distributions for individual pathways of interest. As an example, consider the Bock dataset, where the significant KEGG pathways for spliceosome, oxidative phosphorylation and ECM-receptor interaction each cover different aspects of hESC regulation. For both the spliceosome and oxidative phosphorylation pathways, greater transcriptional stability was manifested through a significantly higher number of low variability genes, in addition to significantly fewer medium variability genes compared to the reference distribution (Figs. 3A–3C). A significant reduction in genes with high variability in the ESCs was also observed for the oxidative phosphorylation pathway compared to the reference (Fig. 3C). The opposite trend was observed for genes in the ECM-receptor interaction where there was a significant increase in genes with medium, high and very high levels of expression variability (Fig. 3D). This pathway also had a concurrent reduction in low variability genes compared to the reference distribution.

## Highlighting differences between ESC and iPSC usage of global gene expression programs using *pathVar*

Human iPSCs were also profiled by Bock using microarrays, and we use this data to investigate how ESCs and iPSCs differ with respect to expression variability. *pathVar* identified five KEGG and thirty REACTOME statistically significant pathways between the ESCs and iPSCs (Table S5, $P$-value < 0.01). The significant KEGG pathways reflected aberrant gene expression activity in ribosome, oxidative phosphorylation, DNA replication and disease processes (Huntington's disease and Parkinson's disease, Table S6). The significant REACTOME terms were associated with cell cycle, splicing, and metabolic
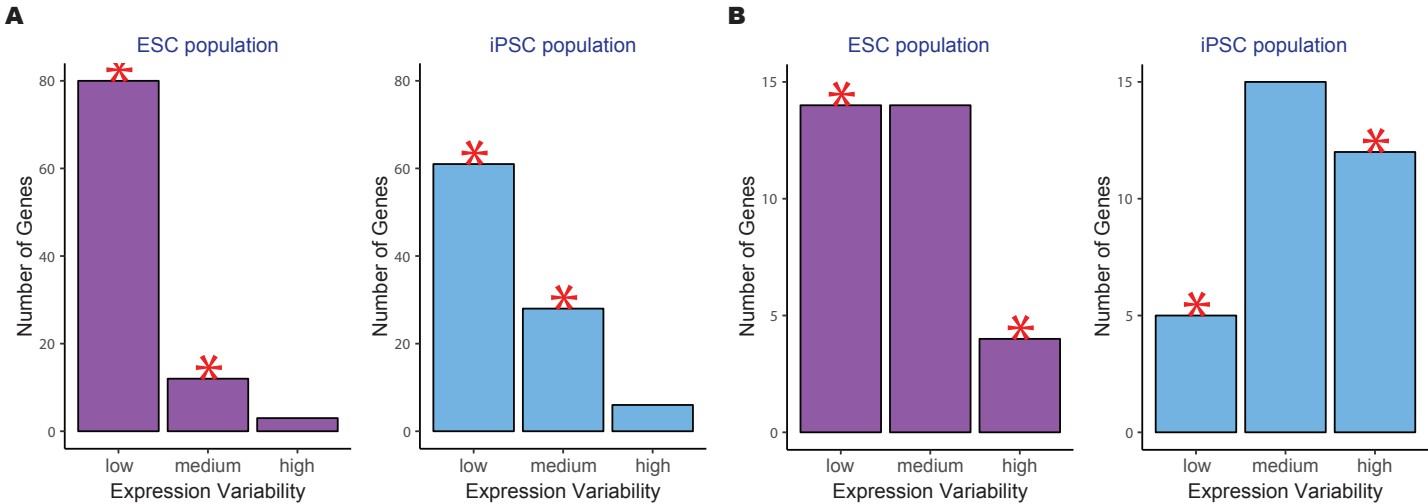

**Figure 4** **Example of two significant KEGG pathways when comparing human embryonic stem cells (ESC) and induced pluripotent stem cell (iPSC) data using the two-group *pathVar* analysis.** (A) Oxidative phosphorylation (hsa00190), (B) DNA replication (hsa03030). In both pathways, a higher number of genes with lower variability are present in ESCs versus iPSCs. The red stars indicate a significant difference between the two groups for a specific level of expression variability.

processes as well as DNA replication and repair, namely homologous recombination pathways (Table S6).

We then compared the variability count distributions in ESCs versus iPSCs for two significant KEGG pathways, oxidative phosphorylation and DNA replication (Fig. 4). Both pathways showed increased gene expression stability in ESCs compared to iPSCs. For the oxidative phosphorylation pathway, a significantly higher number of low variability genes were in ESCs versus iPSCs. The same trend was observed for the DNA replication pathway where there was a significant reduction in the number of highly variable genes in ESCs compared to iPSCs.

## Benchmarking *pathVar* results using gene expression variability versus average expression and two different GSEA approaches

*pathVar* results were benchmarked against those obtained using an average expression statistic to investigate the utility of expression variability analyses. Under this average-based setting, genes were first classified into discrete levels using average expression, corresponding to low, medium, and high levels of absolute expression. This average-based implementation allows for direct comparison with the results that are obtained when studying gene expression variability by substituting the sample mean directly in place of the variability statistic. In this way, we aim to distinguish the pathways that are identified as having changes in expression variability that are not detected by changes in average expression. Four statistically significant pathways ($P$-value $< 0.01$, Table S7) were identified as having differences in average expression between iPSCs and ESCs (three REACTOME terms: Heme biosynthesis, Sphingolipid metabolism, Metabolism of porphyrins; and one KEGG pathway African *trypanosomiasis*). These results do not seem very informative or relevant to stem cell regulation. This could be due to the over-simplified nature in which the

average-based comparison was performed and suggested that this benchmarking approach could be improved.

We next evaluated the performance of *pathVar* against two versions of GSEA. The first method was based on generating *P*-values from limma (*Ritchie et al., 2015*) to determine differential expression between the two groups being compared. The *P*-values for genes belonging to the same pathway were then used as input to a Kolmogorov–Smirnov (KS) test which assessed whether the distribution of this set of *P*-values was identical between the two groups. Since the KS test is affected by tied values, we used the bootstrap version of the KS test (R function ks.boot). This version of GSEA is similar to the implementation of (*Mootha et al. (2003)*. The second GSEA method is a stand-alone tool by *Oron, Jiang & Gentleman (2008)* which uses a linear model to assess the differential usage of a pathway or gene set and calculates permutation-based *P*-values to determine the significance of each pathway. The resulting set of *P*-values for all pathways was adjusted for multiple testing using the Benjamini–Hochberg method (*Benjamini & Hochberg, 1995*).

In general, the two GSEA approaches detected significant pathways that had a moderate to high degree of overlap with the significant pathways that were identified by *pathVar* for all three of the cancer-based comparisons, and the mouse hippocampus versus striatum comparison. This suggests that there are significant pathways common to both GSEA and *pathVar*, but the latter also identifies distinct pathways since the highest overlap for all six comparisons for GSEAlm was 61.1% (KEGG), 96.2% (REACTOME), and for GSEA using limma *P*-values was 22.2% (KEGG), 33.3% (REACTOME). There are a considerable number of pathways identified uniquely by *pathVar*, suggesting that there is value in using multiple approaches to determine differential pathway usage or expression.

## Regulatory insights from other datasets confirm utility of looking at pathways with changes in both gene expression variability and average gene expression using pathVar and GSEA

Examples using stem cells illustrate how *pathVar* works in practice; however, the method can be applied to virtually any gene expression dataset. To highlight the generalizability of *pathVar*, we selected ten other datasets that cover a variety of biological and experimental variables. Collectively, these ten datasets were generated from multiple technology platforms that featured samples from human, mouse and parasite which represent a range of different disease phenotypes (see Text S3).

Three cancer RNA-seq datasets from the Cancer Genome Atlas (TCGA) were selected; these were the ovarian serous cystadenocarcinoma (OVC) (*Cancer Genome Atlas Research, 2011*), acute myeloid leukemia (AML) (*Cancer Genome Atlas Research, 2013*), and glioblastoma multiforme (GBM) (*Brennan et al., 2013*) cohorts. An infectious disease was included where transcriptomes from patients infected with cerebral malaria were profiled using microarrays (*Daily et al., 2007*), as well as the *Plasmodium falciparum* parasites that the patients were infected with (*Feintuch et al., 2016*). A genetic disorder was featured where patient-derived iPSCs were collected from Down syndrome (DS) donors and profiled using microarrays, with a set of matched controls from healthy subjects (*Briggs et al., 2013*). A microarray dataset from a normal human population via

the Geuvadis study (1000 Genomes Project) was used (*Genomes Project et al., 2010*), as well as two mouse datasets that profiled tissues from two different regions of the brain, the hippocampus and the striatum using microarrays (*Park et al., 2011*). *pathVar* identified statistically significant pathways from KEGG and REACTOME pathway terms for the ten different one-group analyses (Table S8), and five independent two-group comparisons (Table S9) of all ten datasets.

The results from the different analyses were used to investigate the uniqueness of analyses based on GSEA versus gene expression variability. The number of significant pathways that had changes in both mean and variability via GSEA and *pathVar* respectively showed that for all cases, the amount of overlap in significant pathways differed depending on the datasets that were used. This suggests that average and variability-based statistics reflect different ways in which cells may use their transcription programs depending on the biological context (Tables S8 and S9). It is interesting to note that for the KEGG pathways, *pathVar* results for the DS versus WT iPSC comparison, and the mouse hippocampus versus striatum comparison both had zero overlap between average-based and variability-based significant pathways (Table S9A). In fact, both comparisons also yielded no significant pathways with a difference in average, whereas pathways were found to have a difference in gene expression variability. These two comparisons are extreme examples where the analyses of gene expression variability identify changes in the transcriptional program, whereas average-based analyses do not yield significant results.

Overall, it was apparent that the transcriptional features responsible for distinguishing one phenotype from another are exerted through changes in average expression or variability in expression for key pathways. To further investigate the relevance of these different modes, we focused only on the ten most significant pathways from the *pathVar* results obtained for the three cancer versus normal comparisons (Tables S10 and S11). For all three cancers, the KEGG DNA replication pathway and REACTOME "DNA strand elongation term" had significant changes in both average and variability of expression. Other terms with changes in both average and variability were related to DNA damage response pathways, such as "base excision repair" (Table S10) for AML versus normal, and "non-homologous end-joining" for OVC versus normal (Table S11).

Five KEGG pathways had changes in variability only, that were consistent in all three cancer comparisons; these were the pathways involved in Epstein-Bar virus infection, cell cycle, Fanconi anemia, lysosome and apoptosis (Table S10). The Epstein-Bar virus is associated with certain kinds of cancer like lymphoma or carcinoma. Apoptosis is also an important pathway for tumors because its inactivation is central in the development of cancer. Similarly, for the REACTOME terms, those unique to changes in variability were related to DNA repair and replication (SLBP dependent processing of replication-dependent histone pre-mRNAs) for the AML and GBM comparisons. For OVC, several terms were related to the cell cycle, e.g., G1 phase, cyclin D associated events in G1, cyclin A/B1 associated events during G2/M transition (Table S11C).

## DISCUSSION

With pathway-centric approaches like GSEA and OR now such ubiquitous features of transcriptomic analyses, *pathVar* represents a natural adjunct to this kind of analysis. Our results from analyses of ESCs and other datasets have demonstrated that it is not uncommon for phenotypes to be regulated by pathways that have altered levels in both average expression and expression variability, as well as pathways unique to either statistic. Therefore, to derive more accurate insights into transcriptional control, our results suggest that pathway-based analyses should include the detection of changes in both population statistics. *pathVar* may also be used to investigate the regulatory control associated with common targets of transcription factors (*Lachmann et al., 2010*; *Matys et al., 2006*), microRNA (*Chou et al., 2016*; *Wong & Wang, 2015*), or lncRNAs (*Jiang et al., 2015*; *Quek et al., 2015*), genes with common variants identified from genome-wide association studies (*Welter et al., 2014*), or other regulatory features (*Guo et al., 2016*) that may benefit from further study of gene expression variability patterns.

In the analysis of ESCs versus iPSCs, *pathVar* identified very few significant pathways relative to the other two-group comparisons conducted (Table S9). This result likely reflects the high degree of similarity that exists between iPSC and ESC transcriptional programs. While the two cell populations have identical developmental capabilities, in some instances, iPSCs retain a limited memory of the gene expression program of the cell of origin. Some of the significant pathways identified by *pathVar* may point to different usage of metabolic processes or the cell cycle by iPSCs and ESCs. Overall, we see more variability in the iPSCs than the ESCs and the increased heterogeneity for these pathways could reflect underlying differences due to donor variability, or experimental factors associated with their generation.

Of the six two-group comparisons performed, it is interesting to note that the ESC and iPSC comparison also had the least number of significant pathways (Table S9) and this may have been due to the fact that all other comparisons were between a disease and normal group, or in the case of the mouse data, between two distinct regions of the brain (Text S3). This result suggests that the degree of perturbation to a transcriptome in the presence of a tumor, or extra chromosome, or even a different anatomical region of the same organ, is greater globally, than how iPSCs differ from ESCs.

The observation that pathways were significant for changes in both average expression and gene expression variability, as well as those identified by GSEA, reflects the different modes in which cells are using pathways to regulate transcriptional signals. For the cancer-based comparisons, common themes were observed across cancer types where pathways involved in DNA replication and DNA damage response had significant changes in average and variability (AML versus 1000 Genomes, OVC versus 1000 Genomes, Tables S10 and S11). The reliance of DNA replication pathways may be to facilitate the proliferative nature of tumor cells, while the pathways that control DNA damage response are important for tumor cells to remain viable in the presence of increased rates of mutation. This result suggests that a critical factor to understanding how cancer subverts cellular pathways to promote growth and evade apoptosis more accurately may lie in focusing on how gene

expression is being regulated based on average expression and expression variability from cell to cell, or from patient to patient.

For the different datasets that were analyzed, the degree of dependency observed between the variability statistic and the average expression varied. This was one criterion that we suggest be used to select the variability statistic in *pathVar* where a measure with the least dependence with average expression is preferred. Nevertheless, some kind of dependency is bound to occur, and this is likely to affect the overlap in the number of significant pathways detected by *pathVar* and also other approaches, such as GSEA. To investigate this effect further, we looked at the Pearson correlation coefficients between average expression and expression variability and the number of significant pathways identified between *pathVar*, and both GSEA implementations (Table S12). Although only six two-group comparisons were available to investigate this effect, we saw that four comparisons where the overall data had positive correlations between average and variability had higher numbers of overlapping pathways than the two comparisons with overall data that had negative correlations. More datasets are needed to confirm this observation though, as two of the positive correlations were also small.

Single cell heterogeneity, or inter-cellular variation is a common reality of all cell populations since even isogenic cells have some degree of stochastic gene expression. Across the transcriptome, gene expression variability is not distributed uniformly, and its functional contribution of transcriptional regulation at the single cell-level remains largely unknown. Genes with decreased variability may be useful as potential markers since they have a higher degree of generalizability, where it is easier to predict the expression state for such a gene in any cell in the population. Although the *pathVar* method is applicable for both single cell and bulk cell datasets, the interpretation of gene expression variability in the context of single cells would provide even more precise insights into how cells are controlled by the transcriptional regulation of certain pathways.

Our analyses on ESCs allowed the opportunity to investigate how expression variability differed between bulk and single cell populations (Text S4). The overall distributions of expression variability indicated that the transcriptome in the bulk ESC data was characterized by lower levels of expression variability compared to both sets of single cells where more genes had medium to higher expression variability. The dependency between variability and average expression appeared to be stronger in the single cell data, while for bulk data, the two variables were almost independently-distributed. It is worthwhile to highlight that the bulk data was profiled using a microarray platform, and the single cell data was from RNA-sequencing so the effect of technology could also be contributing to these observed differences. Nonetheless, these comparisons underscore the importance of investigating the biological significance of variability at single cell resolution.

## CONCLUSION

The *pathVar* method identifies pathways with aberrant distributions in gene expression variability relative to either a reference distribution, or a contrasting control group. The method is based on an intuitive framework where either a multinomial exact test or

Chi-squared test is employed to assess the differences in variability distributions for each pathway using definitions from any standard or custom annotation system. A Binomial test is then used to identify genes within a specific pathway that show differences in gene expression variability. Comparisons benchmarking results from *pathVar* applied to a variety of gene expression datasets against those obtained using GSEA identified significant pathways showing changes either in average expression, expression variability or both. These results indicate that both population statistics are useful for interpreting significant alterations of pathways and gene sets that underlie transcriptional regulation. The implications of these results suggest that future studies may benefit from analyses of gene expression variability to complement standard analyses of average expression.

## ACKNOWLEDGEMENTS

We thank Drs. Barbra Birshtein, Maureen Charron, and Deepa Rastogi for valuable feedback. We also thank Raymund Bueno for testing the method.

### Funding
SZ and JCM were supported by the New York State Department of Health (NYSTEM Program) shared facility grant (C029154). The funders had no role in study design, data collection and analysis, decision to publish, or preparation of the manuscript.

### Grant Disclosures
The following grant information was disclosed by the authors:
New York State Department of Health (NYSTEM Program): C029154.

### Competing Interests
The authors declare there are no competing interests.

### Author Contributions

- Laurence de Torrente and Jessica C. Mar conceived and designed the experiments, performed the experiments, analyzed the data, wrote the paper, prepared figures and/or tables, reviewed drafts of the paper.
- Samuel Zimmerman performed the experiments, analyzed the data, prepared figures and/or tables, reviewed drafts of the paper.
- Deanne Taylor and Christine A. Wells reviewed drafts of the paper, provided feedback that improved the method during its development.
- Yu Hasegawa analyzed the data, prepared figures and/or tables.

### Data Availability
All raw data and code is available at GitHub:

https://github.com/jessicamar/pathVar

The software is publicly available as an open source package from Bioconductor:

https://bioconductor.org/packages/release/bioc/html/pathVar.html

## Supplemental Information

Supplemental information for this article can be found online at http://dx.doi.org/10.7717/peerj.3334#supplemental-information.

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
