# Peer review of "pathVar: a new method for pathway-based interpretation of gene expression variability"

_PeerJ, doi:10.7717/peerj.3334_

## Round 0.1 · original submission · Major Revisions

The manuscript has been evaluated by two independent reviewers. They both had a similar reservation regarding the advantage of your approach relative to comparable and existing alternatives.
Also the testing of the sensitivity of the method suggested by the second reviewer could make the current manuscript much more convincing.

These are clearly major revisions, that to my mind cannot be taken lightly and should be performed in a meticulous fashion. Only if you manage to do this I would be happy to re-evaluate a thoroughly revised version.

Reviewer 1 ·

Basic reporting

No comments

Experimental design

No comments

Validity of the findings

No comments

Additional comments

This paper proposes a significance test that compares a categorical distribution of genes of a given pathway to a reference distribution. The categorical distribution is based on a classification of genes based on their standard deviation (SD) of expression levels, either using a subdivision into three equally sized groups, or a fitted normal mixture model. The significance test is a chi-squared test based on a multinomial sampling distribution. It comes in two variants, i.e. an exact test and an asymptotic test.

The proposed test is a standard procedure that is available in any decent statistical package. As such, the proposed statistical method is fine and the illustration is useful. I do not see, however, how this can justify a full scientific paper that is focused on the methodology alone. I would rather expect this procedure to be applied in a collaborative project where the authors provide the statistical analysis.

·

Basic reporting

The authors present a computational method for detecting pathways with significant differences in gene expression variability either between two experimental conditions, or compared to the background in a single experimental condition. The paper is well written and software source code has been made available.

Experimental design

Overall the design of the method validation experiments is appropriate. However there are 3 topics where more details/additional experiments would be welcome:

(1) The effect of sample size on the method/results is not analyzed. For instance, in the ESC analysis, standard deviations are estimated from respectively 20, 8 and 26 samples. Knowing that the relative standard error of the standard deviation is ~1/sqrt(n-1), with n the sample size, these errors will be quite large, potentially leading to misclassifying the variability levels of many genes. I recommend testing the effect of sample size in the power calculation simulations and/or by subsampling from a larger dataset.

(2) Intuitively I would expect variability to play a more important role when analyzing single-cell data (where variability more likely has a biological origin, at least for medium to highly expressed genes) compared to bulk transcriptome data (where it is more likely to have a technical origin). The authors have analyzed single-cell as well as bulk data, so it would be interesting to comment on any consistent differences between them.

(3) A critical question is whether "differential variability" measures something different than "differential expression". Variability is generally higher for lowly expressed genes, and thus one expects that at least part of the differential variability can be explained by differential expression. The authors recognize this issue by comparing their method to one based on average expression. However their choice of method really is overly simplistic: they bin genes according to average expression and then test for differences between bin counts. I am not aware of this being a commonly accepted method for differential expression testing (it certainly isn't used in the GSEA paper cited), and, as the authors note, its results don't make much sense in the iPSC vs ESC comparison. It is essential in my opinion that pathVar be compared against a method where standard differential expression p-values obtained by a method such as limma are used as input for the cited GSEA method. Additionally it would be informative if the authors could discuss the results in the context of the above comment, where a certain amount of overlap is expected due to the dependence of the SD on the average expression.

Validity of the findings

See comments on Experimental Design.

Additional comments

No additional comments.

---

## Round 0.2 · accepted · Accept

My perception is that the revised manuscript has sufficiently addressed the questions raised by the reviewers. That this has increased the quality of the manuscript, so that it is now acceptable for publications. Congratulations!

Reviewer 1 ·

Basic reporting

In their response to my comments, the authors give good reasons justifying the publication of this paper despite the fact that it uses standard statistical tests. I think this justification should be given somewhere in the paper, either in the introduction or the discussion.

Experimental design

The authors study power of their procedure. In isolation, this power analysis is of little use. The power analysis should include the two alternative tests the authors also apply to their data.

Validity of the findings

no comment

Additional comments

no comment

·

Basic reporting

no comment

Experimental design

no comment

Validity of the findings

no comment

Additional comments

The authors have addressed my previous comments.